

# Can the triglyceride-glucose index predict the risk of stroke? A meta-analysis of high-quality studies with 12.8 million participants

Gang Xin, Huiya Li and Ji Jiang

Emergency Department, The Affiliated People's Hospital of Ningbo University, Ningbo, Zhejiang, China

## ABSTRACT

**Objective**. The triglyceride-glucose index (TyG) has been actively researched for predicting several diseases. However, high-quality evidence assessing its ability to predict stroke is lacking. We conducted a meta-analysis of high-quality studies examining if TyG can predict stroke in the general population.

**Methods**. Embase, PubMed, CENTRAL, Web of Science, and Scopus databases were searched until 13th January 2025. Cohort studies on the general population, excluding those with baseline stroke or cardiovascular disease, with a minimum follow-up of four years and reporting an adjusted association between TyG and stroke were included. TyG was assessed as both a categorical and continuous variable.

**Results**. A total of 13 studies with 12,898,434 individuals were eligible. The overall incidence of stroke was 0.89%. Meta-analysis indicated a statistically significant increased risk of stroke between higher *vs* lower values of TyG (risk ratio (RR): 1.27 95% confidence interval (CI) [1.19–1.35] $I^2 = 66\%$). Per unit increase in TyG was also associated with a statistically significant increase in the risk of stroke (RR: 1.16 95% CI [1.07–1.27] $I^2 = 89\%$). Most results remained unchanged on subgroup analysis based on location, excluded population, stroke diagnosis, TyG data, and follow-up. Meta-regression using moderators sample size, age, male gender, diabetes mellitus, hypertension, TyG cut-off, stroke incidence, and follow-up also failed to reveal significant results.

**Conclusion**. High TyG is associated with increased risk of stroke in the general population.

Corresponding author
Gang Xin, xg00142024@163.com

## INTRODUCTION

Stroke is now the second major cause of mortality worldwide leading to about 1/3rd of all disabilities (*Lozano et al., 2012*). Statistics indicate that about 13 million new incident cases of stroke were diagnosed in 2016 with about 87% being ischemic stroke (*Saini, Guada & Yavagal, 2021*). Chinese data from 2020 shows that approximately 3.4 million patients were diagnosed with first-ever stroke causing about 2.3 million deaths (*Tu et al., 2023*). About 76% of strokes develop in individuals without a prior history of the disease (*Saver et al., 2015*). The illness is a life-altering event for those who experience the disease as well as

for families and caregivers. About 26% of elderly experiencing stroke become dependent on activities of daily living and about 46% have cognitive impairment (*Go et al., 2014*). These findings support the fact that active and effective prevention can help in reducing the disease burden and there is an urgent need for scaling up the primary prevention programs (*Lozano et al., 2012*). In this context, the development of accurate risk prediction markers and models can help identify high-risk individuals who can be monitored and targeted by effective interventions to reduce the risk of stroke (*Xu et al., 2021*). Despite the establishment of several stroke prediction models and biomarkers in the past few decades, researchers have been unable to identify a single model or marker that is highly effective in predicting the risk of stroke (*Xu et al., 2021*; *Lu et al., 2021*; *Lip et al., 2022*; *Ihle-Hansen et al., 2023*).

Research indicates that insulin resistance could have a major role in the pathogenesis of stroke (*Ding et al., 2022*). Insulin acts as a protective agent for the brain by preventing ischemia, oxidative stress, and apoptosis-induced brain tissue damage. It also modulates cholesterol metabolism in neural tissues and astrocytes and is known to improve cognitive dysfunction in Alzheimer's disease (*Agrawal et al., 2021*; *Ding et al., 2022*). Increased insulin resistance has been linked with a higher risk of stroke in the general population (*Zheng et al., 2024*). However, the current gold standard for assessing insulin resistance, *i.e.,* the hyperinsulinemic-euglycemic clamp is too complex and expensive to be applied routinely in clinical practice (*Cersosimo et al., 2014*). A more accessible marker can be the homeostasis model assessment of insulin resistance (HOMA-IR) index (*Matthews et al., 1985*). Nevertheless, its routine clinical application is also not economical and convenient. The triglyceride-glucose (TyG) index, is considered to be a biomarker for insulin resistance which is calculated using the formula: fasting triglycerides (mg/dl)×fasting blood glucose (mg/dl)/2 (*Liao et al., 2022*). Studies have shown that high TyG levels are significantly associated with increased risk of coronary artery disease (CAD), contrast-induced nephropathy, hypertension (HT), diabetes mellitus (DM), atrial fibrillation, metabolic dysfunction associated fatty liver disease, metabolic syndrome, and stroke. Moreover, a positive association has also been demonstrated between high TyG and the prognosis of CAD and stroke (*Yin et al., 2024*; *Nayak et al., 2024*).

On the question of its predictive ability for stroke, three prior meta-analysis studies (*Liao et al., 2022*; *Feng et al., 2022*; *Yang et al., 2023*) with eight to eleven studies each have shown that high TyG may be a potential marker for stroke. However, these reviews have several limitations including inclusion of studies on specific populations (like HT, DM, CAD), use of cross-sectional data, and studies with overlapping data. Moreover, the low number of studies with a small number of participants is an additional hindrance that limits the acceptability of the evidence. To overcome the limitations of prior reviews and to present the best possible evidence in the literature, we conducted an updated meta-analysis including only high-quality cohort studies to assess the ability of TyG to predict stroke in the general population.

## MATERIALS AND METHODS

### Registration

Before beginning the study, all reviewers formulated a protocol which was registered and outlined on PROSPERO. The registration number was CRD42025636156. The review is presented as per the guidelines of PRISMA (*Page et al., 2021*). Ethical approval was not needed as the study was based on published literature.

### Data sources and searches

Databases of Embase, PubMed, CENTRAL, Web of Science, and Scopus were searched for all observational studies evaluating the ability of TyG to predict stroke. Following Medical Subject Headings (MeSH) and free keywords were used: 'triglyceride-glucose index', 'triglyceride and glucose index', 'TyG index', 'triglyceride glucose index', 'triacylglycerol glucose index', 'Stroke', 'Cerebrovascular Accident', 'CVA', 'Brain Vascular Accident', and 'Cerebrovascular Disease'. Detailed search strategies for all databases can be found in Table S1. The bibliography for potential articles meeting the inclusion criteria and past reviews was also manually examined. Lastly, a supplemental search was run on Google Scholar for any other potential articles in gray literature. Two reviewers (GX, HL) independently performed the search which was last updated on 13th January 2025.

### Eligibility criteria

A detailed criteria to include only high-quality studies was formulated by the reviewers. Studies were included in the review provided that (1) They were cohort studies conducted on the general population without a prior history of stroke or cardiovascular disease (CVD). (2) They assessed the temporal association between baseline TyG measurements and the risk of stroke. (3) They reported outcomes as a multiple covariate-adjusted effect size. (4) Mean or median follow-up was at least four years.

Exclusion criteria were: (1) Studies not reporting independent data on stroke. (2) Study on a cohort with a prespecified illness like DM, HT, CAD, *etc.* (3) Studies using the same database with overlapping study periods. In such cases, the study fulfilling the above-mentioned criteria and with the largest sample size was chosen. We also did not include articles only in abstract form, thesis, and editorials.

### Study selection

The search queries were run on respective databases and all results were collated and deduplicated in EndNote software (version X9.3.3, Thomson Reuters, Philadelphia, PA, USA). The remaining studies were analyzed for eligibility by examining the titles and abstracts. Studies chosen for further analysis by either reviewer (GX, HL) were downloaded full texts were assessed. The final selection was after the agreement of both reviewers. All disagreement was resolved through discussion with the third reviewer (JJ).

### Data management

Two reviewers (GX, HL) extracted information regarding the author, publication year, population included, exclusion of stroke or CVD, demographic details and comorbidities, identification of stroke, stroke incidence, TyG data (categorical, continuous), cut-off,

adjusted covariates, follow-up, and outcomes. Most studies segregated TyG data as quartiles or tertiles comparing the highest with the lowest groups. Data on TyG as a continuous outcome was also extracted for the meta-analysis. If several adjusted models of the outcome were reported by the studies, the model with maximum adjustment was selected. We did not make any assumptions about missing data. The corresponding author of the study was to be contacted in such cases and case of no response, the study was to be omitted from the meta-analysis.

### Risk of bias

The quality of studies was assessed using the Newcastle Ottawa Scale (NOS) (*Wells et al., 2020*). Two reviewers (GX, HL) participated in the assessment, and disagreements were resolved by consensus. The reviewers were not blinded. Each study was judged on the following domains using the prespecified questions of NOS: participant selection, group comparability, and outcomes. A score of 8–9 indicated high, 6–7 indicated medium, and <6 indicated low quality. The reviewer JJ was involved to resolve conflicts.

### Statistical analysis

The software used was "Review Manager" (RevMan, version 5.3) for the primary meta-analysis. We pooled data of TyG as a categorical and continuous variable separately in an inverse variance random-effects meta-analysis model. A random effects model was chosen due to the expected baseline heterogeneity between the studies which were from different countries and variable populations. Results were generated as risk ratio (RR) and 95% confidence intervals (CI). Heterogeneity among studies was assessed through Cochran's Q statistic and the $I^2$ index. $I^2$ of over 50% and/or $P < 0.05$ indicated significant heterogeneity. The influence of individual studies was judged by sensitivity analysis which was done in the Review Manager software itself. One study at a time was removed from the meta-analysis to assess the stability of the results. Publication bias was checked using funnel plots and the Egger's test.

Assessment of the source of heterogeneity was by subgroup and meta-regression analysis. The latter was conducted using the Meta-Essentials tool (*Suurmond, Van Rhee & Hak, 2017*). Studies were divided based on location (Chinese, Asian, Western), excluded population (all CVD or all stroke), stroke diagnosis (ICD codes, medical records, physician-diagnosed), TyG data (quartile or tertile), and follow-up ($\geq$10 or <10 years). Moderators were all continuous variables namely, sample size, age, male gender, DM, HT, TyG cut-off, stroke incidence, and follow-up.

## RESULTS

### Search results

We have presented the search results in Fig. 1. Of the 3,400 studies found from all databases, we removed 2,488 duplicates. A total of 1,122 studies underwent meticulous screening by the reviewers. Jointly, they selected 40 studies for further analysis. After full-text reading, 13 were selected for the review (*Hong, Han & Park, 2020*; *Zhao et al., 2021*; *Cho et al., 2022*; *Liu et al., 2022b*; *Liu et al., 2022a*; *Che et al., 2023*; *Wang et al., 2023*; *Muhammad et al.,*
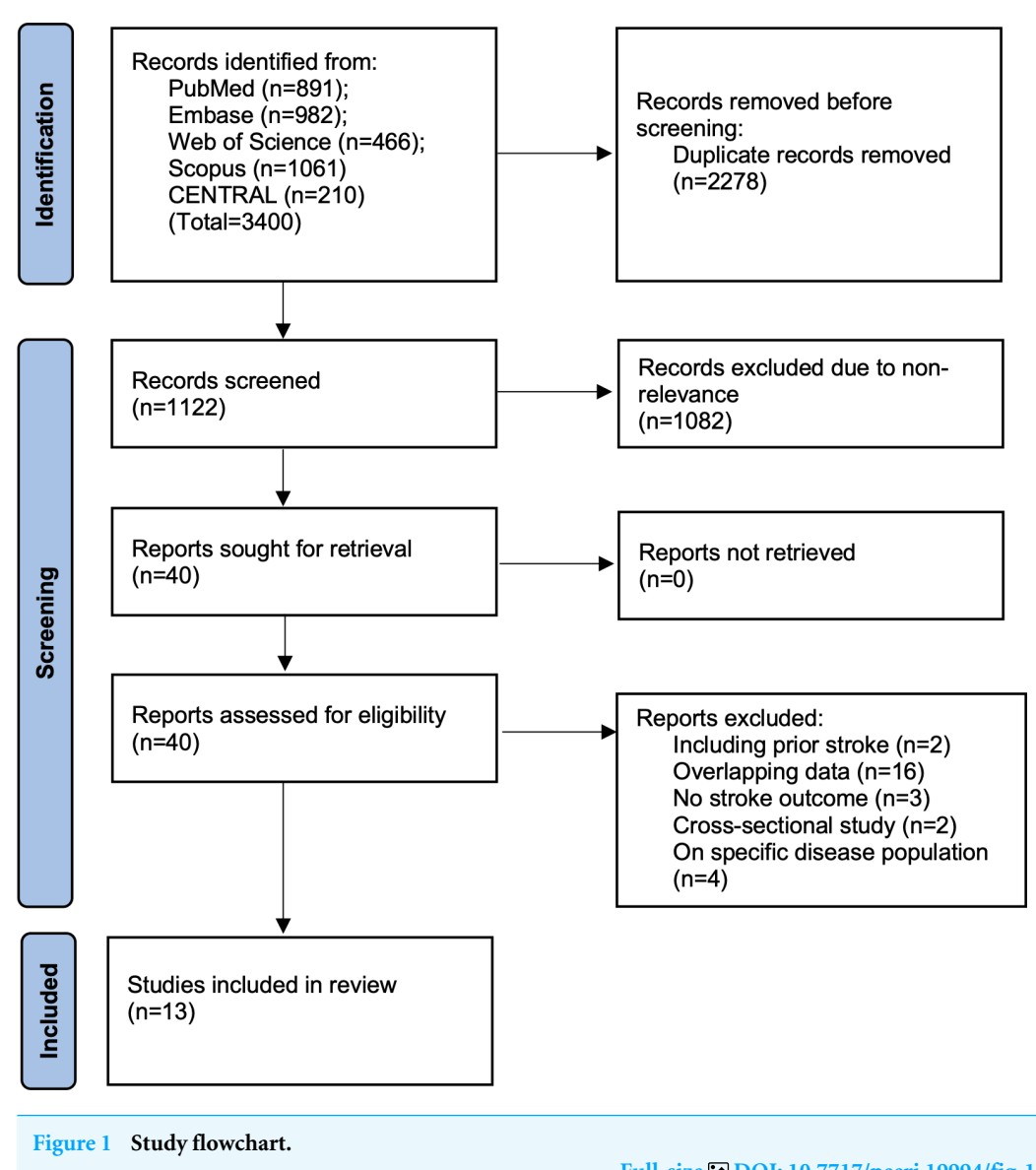

**Figure 1   Study flowchart.**

*2023*; *Wan et al., 2023*; *Yao et al., 2024*; *Li et al., 2024a*; *Li et al., 2024b*; *Rafiee et al., 2024*). Agreement between reviewers was high (kappa = 0.95). The search of additional sources did not reveal any missed study. A list of excluded studies can be found in Table S2.

## Study details

As noted in Table 1, the majority of studies were from China utilizing different databases. Two studies were from Korea from the same database but with non-overlapping age of the sample (one was >40 years and the other 20–39 years). One study each was available from the UK, USA, Sweden, and Iran. Ten studies excluded all known CVD patients (including stroke) while three excluded only prior stroke patients. In total, 12,898,434 individuals were enrolled in the 13 studies. The incidence of stroke varied from 0.13 to 13.3%. Overall, the incidence of stroke was found to be 0.89% (115,068/12,989,434). Two studies excluded

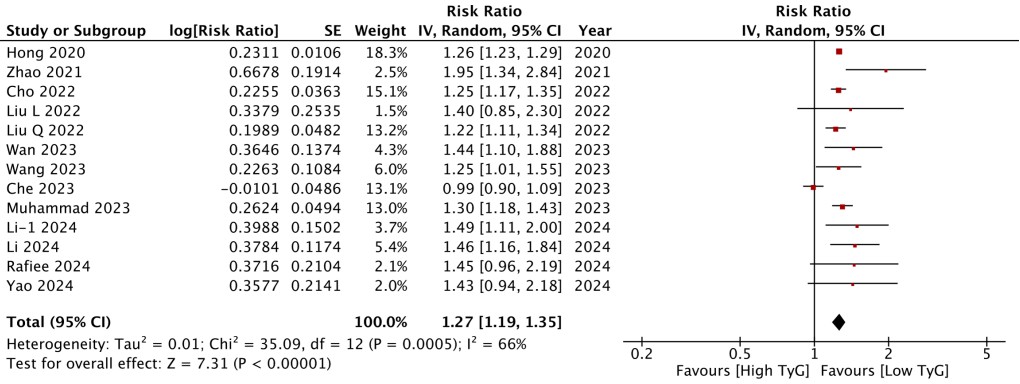

| Study or Subgroup | log[Risk Ratio] | SE | Weight | Risk Ratio IV, Random, 95% CI | Year |
|---|---|---|---|---|---|
| Hong 2020 | 0.2311 | 0.0106 | 18.3% | 1.26 [1.23, 1.29] | 2020 |
| Zhao 2021 | 0.6678 | 0.1914 | 2.5% | 1.95 [1.34, 2.84] | 2021 |
| Cho 2022 | 0.2255 | 0.0363 | 15.1% | 1.25 [1.17, 1.35] | 2022 |
| Liu L 2022 | 0.3379 | 0.2535 | 1.5% | 1.40 [0.85, 2.30] | 2022 |
| Liu Q 2022 | 0.1989 | 0.0482 | 13.2% | 1.22 [1.11, 1.34] | 2022 |
| Wan 2023 | 0.3646 | 0.1374 | 4.3% | 1.44 [1.10, 1.88] | 2023 |
| Wang 2023 | 0.2263 | 0.1084 | 6.0% | 1.25 [1.01, 1.55] | 2023 |
| Che 2023 | −0.0101 | 0.0486 | 13.1% | 0.99 [0.90, 1.09] | 2023 |
| Muhammad 2023 | 0.2624 | 0.0494 | 13.0% | 1.30 [1.18, 1.43] | 2023 |
| Li−1 2024 | 0.3988 | 0.1502 | 3.7% | 1.49 [1.11, 2.00] | 2024 |
| Li 2024 | 0.3784 | 0.1174 | 5.4% | 1.46 [1.16, 1.84] | 2024 |
| Rafiee 2024 | 0.3716 | 0.2104 | 2.1% | 1.45 [0.96, 2.19] | 2024 |
| Yao 2024 | 0.3577 | 0.2141 | 2.0% | 1.43 [0.94, 2.18] | 2024 |
| **Total (95% CI)** | | | **100.0%** | **1.27 [1.19, 1.35]** | |

Heterogeneity: Tau² = 0.01; Chi² = 35.09, df = 12 (P = 0.0005); I² = 66%
Test for overall effect: Z = 7.31 (P < 0.00001)

**Figure 2** Meta-analysis of the association between TyG (categorical variable) and risk of stroke. Notes: *Hong, Han & Park, 2020*; *Zhao et al., 2021*; *Cho et al., 2022*; *Liu et al., 2022a*; *Liu et al., 2022b*; *Wan et al., 2023*; *Wang et al., 2023*; *Che et al., 2023*; *Muhammad et al., 2023*; *Li et al., 2024a*; *Li et al., 2024b*; *Rafiee et al., 2024*; *Yao et al., 2024*.

DM patients and in the remaining studies, the prevalence of baseline DM was 2.5–19%. HT ranged from 6.9–69.1%. In studies not excluding baseline CVD, the prevalence was between 5–13.4%. Data on other comorbidities was limited. Most studies used ICD codes or medical records for the identification of stroke. Eleven studies segregated TyG data as quartiles while two presented it as tertiles. Eight studies additionally used TyG as a continuous variable. Adjusted covariates varied between studies. Follow-up ranged from 4.3 to 26.6 years. All studies were high quality and received an NOS score of nine except one which got eight.

## TyG as a categorical variable

Meta-analysis of all 13 studies indicated a statistically significant increased risk of stroke between higher *vs* lower values of TyG (RR: 1.27 95% CI [1.19–1.35]) (Fig. 2). $I^2$ was found to be 66% indicating high heterogeneity. No change in the significance of RR was found by the reviewers during sensitivity analysis. Publication bias was noted on the funnel plot (Fig. 3). Egger's test was not significant ($p = 0.69$). Subgroup analysis showed that the association between TyG and stroke persisted after dividing studies based on location, excluded population, stroke diagnosis, TyG data, and follow-up (Table 2). Heterogeneity was found to be reduced to zero in some of the subgroup analyses like studies on western cohorts, excluding only baseline stroke, using TyG as tertiles, and with follow-up ≥10 years. Meta-regression analysis found that age, male gender, DM, HT, TyG cut-off, stroke incidence, and follow-up did not have a significant effect on the meta-analysis results (Table 3).

## TyG as a continuous variable

Meta-analysis of eight studies showed that a per unit increase in TyG was associated with a statistically significant increase in the risk of stroke (RR: 1.16 95% CI [1.07–1.27]) (Fig. 4). $I^2$ was 89%, indicating high heterogeneity again. No major asymmetry of the funnel plot was noted (Fig. 5). Egger's test was not significant ($p = 0.76$). The RR remained statistically

**Table 1 Details of included studies.**

| Study | Database | Study population | Sample size | Age (y) | Male (%) | DM (%) | HT (%) | CVD (%) | CKD (%) | DL (%) | Stroke diagnosis | TyG data | TyG cut-off^ | Adjusted covariates | Stroke incidence (%) | F/U (y) | NOS score |
|---|---|---|---|---|---|---|---|---|---|---|---|---|---|---|---|---|---|
| Hong, Han & Park (2020) | National Health Insurance Service, Korea | >40y without CVD, not on lipid-lowering or DM medication | 5,593,134 | 52 | 50.5 | 3.7 | 26.9 | 0 | 6.2 | 11.2 | ICD code | Quartiles | NR | Age, sex, smoking, alcohol consumption, regular physical activity, low socioeconomic status, BMI, hypertension, TC, HT medications, warfarin, and aspirin | 1.59 | 8.2 | 9 |
| Zhao et al. (2021) | Rural Chinese Cohort Study | ≥40y without CVD and stroke | 11,777 | 53 | 40.9 | NR | NR | 0 | NR | NR | Clinical and radiological* | Quartiles | 9.14 | Age, gender, marital status, income, education level, smoking, alcohol drinking, physical activity, family history of stroke, HT, resting heart rate, BMI, waist circumference, TC, HDL-C, LDL-C | 5.74 | 6 | 9 |
| Cho et al. (2022) | National Health Insurance Service, Korea | 20–39y without CVD, not on lipid-lowering or DM medication | 6,675,424 | 31 | 59.6 | 0 | 6.9 | 0 | 1.9 | NR | ICD code | Quartiles | 8.34 | Age, sex, BMI, smoking, alcohol consumption, physical activities, income, HT, andTC | 0.13 | 7.4 | 9 |
| Liu et al. (2022a) | Kailuan study, China | Without CVD | 96,541 | 51 | 79.6 | 9 | 43.3 | 0 | NR | 0.75 | WHO criteria | Quartiles | 9.05 | Age, sex, current smoking status, physical activity,education, BMI, HT, DM, HDL-C, LDL-C, hs-CRP, lipid-lowering medication, DM medication, and HTmedication | 5.3 | 10.3 | 9 |
| Liu et al. (2022b) | Eastern China cohort | Without CVD and DM | 6,095 | 48.7 | 49.1 | 0 | 46.5 | 0 | NR | NR | NR | Quartiles | 8.76 | Age, gender, waist-hip ratio, tobacco use, alcohol use, education, physical activity, hypertension, BMI, LDL-C, intake of fat and carbohydrates, use ofantihypertensive drugs, and use of antilipemic drugs | 2.5 | 10.6 | 8 |
| Che et al. (2023) | UK Biobank | 40–69y without CVD | 403,335 | 56.2 | 44.8 | 3.8 | 13.9 | 0 | 2 | 6.7 | ICD code | Quartiles | 9.07 | Age, sex, ethnicity, region, Townsend Deprivation Index, current smoking, physical activity, BMI, HT,TC, LDL-C, uric acid, glycated hemoglobin, estimated glomerular filtration rate, hs-CRP, aspirin, insulin treatment, HT medication, cholesterol-lowering medication, prevalent retinopathy, and CKD | 1 | 8.1 | 9 |
| Muhammad et al. (2023) | Malmö Preventive Project, Sweden | Without stroke | 32,920 | 45 | 67.5 | 2.5 | 5.5 | NR | NR | NR | ICD code | Quartiles | 4.74 | Age, sex, BMI, systolic blood pressure, cholesterol, smoking status, DM, HT medication, physical activity, alcohol | 13.3 | 16.9 | 9 |
| Wan et al. (2023) | Shanghai Suburban Adult Cohort and Biobank, China | Without CVD | 42,651 | 55.7 | 40.3 | 10.2 | 50 | 0 | NR | 34.6 | ICD code | Quartiles | 9.02 | Age, sex, BMI, education level, physical activity, current smoking, current drinking, HDL-C, uric acid,HT medication and DM medication | 1.6 | 4.7 | 9 |

Peerj

**Table 1** (*continued*)

| Study | Database | Study population | Sample size | Age (y) | Male (%) | DM (%) | HT (%) | CVD (%) | CKD (%) | DL (%) | Stroke diagnosis | TyG data | TyG cut-off^ | Adjusted covariates | Stroke incidence (%) | F/U (y) | NOS score |
|---|---|---|---|---|---|---|---|---|---|---|---|---|---|---|---|---|---|
| Wang et al. (2023) | ARIC Study, USA | 45–64y without stroke | 10,132 | 54.1 | 46 | 9 | 33 | 5 | NR | NR | Medical records | Quartiles | NR | Age, race-center, sex, baseline smoking status, alcohol status, BMI, DM, heart failure, and peripheral artery disease, systolic blood pressure, LDL-C, estimated glomerular filtration rate, fibrinogen, lipid-lowering drugs and antihypertensive drugs | 9 | 26.6 | 9 |
| Li et al. (2024a) | Tianjin Brain Study, China | ≥45Y without CVD | 3,534 | 59 | 40.2 | 19 | 69.1 | 0 | NR | NR | Medical records | Tertiles | 9.04 | Sex, age group, smoking status, LDL-C, andhistory of hypertension | 9 | 10 | 9 |
| Li et al. (2024b) | China Health and Retirement Longitudinal Study | Without stroke | 10,569 | 59 | 47.1 | 6.1 | 39.7 | 12.1 | 5.8 | 9.8 | Interview confirming physician-diagnosed stroke | Quartiles | 9.07 | Age, gender, marital status, residence, education level, BMI, smoking status, and drinking status, DM, HT, heart disease, DL, CKD, history of medication use for DM, history of medication use for HT, history ofmedication use for DL, systole blood pressure, diastolic blood pressure, glycated hemoglobin, hsCRP, and estimated glomerular filtration rate. | 7.1 | 7 | 9 |
| Rafiee et al. (2024) | Isfahan Cohort Study, Iran | ≥35y without CVD | 5,432 | 50.7 | 48.8 | 8.4 | 27.8 | 0 | 0 | 87.2 | Physician diagnosis | Tertiles | NR | Age, sex, education, marital status, residency area, global dietary index, smoking status, andtotal daily physical activity, BMI, hypertension, and elevated TC | 3.16 | 11.2 | 9 |
| Yao et al. (2024) | Environment and Chronic Disease in Rural Areas of Heilongjiang, China | ≥35y without stroke, cancer and transient cerebral ischemia | 6,890 | 57 | 38.6 | 5.7 | 53.8 | 13.4 | NR | NR | ICD code | Quartiles | 9.06 | Age, smoking status, drinking status, family history of CVD, family history of stroke, physical activity, TC, HT, DM, CVD | 3.89 | 4.3 | 9 |

**Notes.**

DL, dyslipidemia; DM, diabetes mellitus; HT, hypertension; CVD, cardiovascular disease; CKD, chronic kidney disease; TyG, triglyceride glucose index; ICD, International classification of diseases; F/U, follow-up; y, year; NOS, Newcastle Ottawa scale; HDL-C, high-density lipoprotein cholesterol; LDL-C, low-density lipoprotein cholesterol; BMI, body mass index; TC, total cholesterol; hsCRP, high sensitivity C-reactive protein; NR, not reported.

*Acute focal disturbance within 24 h thought to be due to either intracranial haemorrhage or ischaemia and confirmed by either computed tomography or magnetic resonance imaging.

^For the highest quartile or tertile.

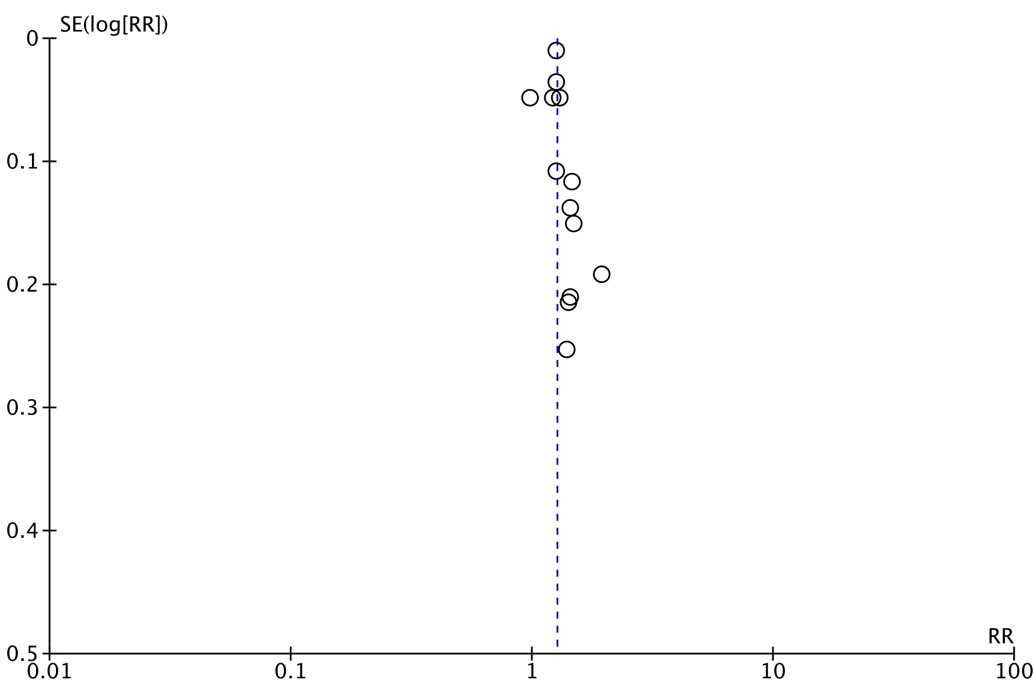

**Figure 3** Funnel plot for the meta-analysis with TyG as a categorical variable.

| Study or Subgroup | log[Risk Ratio] | SE | Weight | Risk Ratio IV, Random, 95% CI | Year |
|---|---|---|---|---|---|
| Liu L 2022 | 0.179 | 0.1355 | 6.5% | 1.20 [0.92, 1.56] | 2022 |
| Liu Q 2022 | 0.1133 | 0.0186 | 16.9% | 1.12 [1.08, 1.16] | 2022 |
| Che 2023 | −0.0202 | 0.0159 | 17.1% | 0.98 [0.95, 1.01] | 2023 |
| Wan 2023 | 0.3293 | 0.0793 | 11.1% | 1.39 [1.19, 1.62] | 2023 |
| Wang 2023 | 0.2784 | 0.0659 | 12.5% | 1.32 [1.16, 1.50] | 2023 |
| Li 2024 | 0.131 | 0.0371 | 15.5% | 1.14 [1.06, 1.23] | 2024 |
| Li-1 2024 | 0.2776 | 0.1024 | 8.9% | 1.32 [1.08, 1.61] | 2024 |
| Yao 2024 | 0.077 | 0.0763 | 11.4% | 1.08 [0.93, 1.25] | 2024 |
| **Total (95% CI)** | | | **100.0%** | **1.16 [1.07, 1.27]** | |

Heterogeneity: $Tau^2 = 0.01$; $Chi^2 = 65.15$, df = 7 (P < 0.00001); $I^2 = 89\%$
Test for overall effect: Z = 3.49 (P = 0.0005)

Favours [High TyG]  Favours [Low TyG]

**Figure 4** Meta-analysis of the association between TyG (continuous variable) and risk of stroke. Notes: *Liu et al., 2022a*; *Liu et al., 2022b*; *Wan et al., 2023*; *Wang et al., 2023*; *Che et al., 2023*; *Li et al., 2024a*; *Li et al., 2024b*; *Yao et al., 2024*.

significant on sensitivity analysis. Subgroup analyses revealed non-significant results for studies on the Western population and those using ICD codes for the identification of stroke. For the other subgroups, the results remained statistically significant (Table 2). On meta-regression analysis (Table 3), only sample size was found to inversely influence the effect size. A larger sample size was associated with a weaker association between TyG and stroke. None of the other moderators were found to be statistically significant.

**Table 2 Subgroup analysis details.**

| Covariates | Groups | Number of studies | Risk ratio [95% confidence intervals] | $I^2$ |
|---|---|---|---|---|
| **TyG categorical variable** | | | | |
| Location | Chinese | 7 | 1.40 [1.24, 1.57] | 32 |
| | All Asian | 11 | 1.27 [1.18, 1.37] | 71 |
| | Western | 2 | 1.29 [1.18, 1.41] | 0 |
| Excluded population | All CVD | 10 | 1.25 [1.16, 1.35] | 72 |
| | All Stroke | 3 | 1.33 [1.22, 1.45] | 0 |
| Stroke diagnosis | ICD codes | 6 | 1.22 [1.12, 1.33] | 81 |
| | Physician diagnosed | 2 | 1.70 [1.28, 2.27] | 8 |
| TyG data | Quartile | 11 | 1.25 [1.17, 1.34] | 70 |
| | Tertile | 2 | 1.48 [1.16, 1.88] | 0 |
| Follow-up | ≥10 years | 6 | 1.27 [1.20, 1.35] | 0 |
| | <10 years | 7 | 1.27 [1.15, 1.41] | 81 |
| **TyG continuous variable** | | | | |
| Location | Chinese | 6 | 1.17 [1.10, 1.24] | 49 |
| | All Asian | 6 | 1.17 [1.10, 1.24] | 49 |
| | Western | 2 | 1.13 [0.84, 1.51] | 95 |
| Excluded population | All CVD | 5 | 1.16 [1.04, 1.30] | 92 |
| | All Stroke | 3 | 1.18 [1.06, 1.30] | 60 |
| Stroke diagnosis | ICD codes | 3 | 1.13 [0.92, 1.38] | 90 |
| | Medical records | 2 | 1.32 [1.18, 1.47] | 0 |
| Follow-up | ≥10 years | 4 | 1.22 [1.09, 1.36] | 63 |
| | <10 years | 4 | 1.12 [0.98, 1.29] | 90 |

**Notes.**
CVD, cardiovascular disease; ICD, international classification of diseases; TyG, Triglyceride glucose index.

## DISCUSSION

A wide range of stroke prediction models and biomarkers have been proposed in the literature to predict the risk of stroke in the general population (*Xu et al., 2021*; *Lu et al., 2021*; *Lip et al., 2022*; *Ihle-Hansen et al., 2023*). There are several routine laboratory markers like albumin, brain natriuretic peptide, serum creatinine, red cell distribution width, total cholesterol, high-density lipoprotein, low-density lipoprotein, and non-high-density lipoprotein cholesterol which have been associated with a higher risk of stroke (*Sughrue et al., 2016*). Other non-routine markers like fibrinogen, E-selectin, interferon-$\gamma$-inducible-protein-10, resistin, and total adiponectin have also been linked with increased risk of stroke (*Prugger et al., 2013*). On the other hand, there are also complex prediction models and machine-learning-based algorithms that claim to accurately predict the risk of stroke (*Lu et al., 2021*; *Lip et al., 2022*). Lastly, specific investigation-based markers like carotid plaque scores are also linked with the risk of stroke (*Ihle-Hansen et al., 2023*). Despite a plethora of research, no single marker has been identified as the gold standard as there are issues of accuracy with simpler laboratory markers and difficulty of application with more complex models.

**Table 3    Details of meta-regression analysis.**

| Variable | Beta | SE | +95% CI | −95% CI | *p*-value |
|---|---|---|---|---|---|
| **TyG categorical variable** | | | | | |
| Sample size | 0.000000005 | 0.000000005 | −0.000000006 | 0.00000002 | 0.34 |
| Age | −0.0008 | 0.002 | −0.005 | 0.003 | 0.64 |
| Male (%) | −0.00002 | 0.001 | −0.003 | 0.002 | 0.99 |
| DM (%) | 0.002 | 0.005 | −0.008 | 0.014 | 0.59 |
| HT (%) | 0.002 | 0.001 | −0.001 | 0.004 | 0.12 |
| TyG cut-off | −0.004 | 0.003 | −0.011 | 0.002 | 0.11 |
| Stroke incidence (%) | 0.007 | 0.004 | −0.001 | 0.015 | 0.66 |
| Follow-up (years) | 0.001 | 0.004 | −0.008 | 0.010 | 0.79 |
| **TyG continuous variable** | | | | | |
| Sample size | −0.0000006 | 0.0000002 | −0.0000011 | −0.0000001 | 0.004 |
| Age | 0.0002217 | 0.0182591 | −0.0429542 | 0.0433976 | 0.99 |
| Male (%) | −0.0024111 | 0.0048529 | −0.0138865 | 0.0090642 | 0.61 |
| DM (%) | 0.0193193 | 0.0088024 | −0.0014951 | 0.0401336 | 0.20 |
| HT (%) | 0.0051778 | 0.0024720 | −0.0006676 | 0.0110231 | 0.40 |
| Stroke incidence (%) | 0.0205690 | 0.0132036 | −0.0106526 | 0.0517907 | 0.12 |
| Follow-up (years) | 0.0071913 | 0.0069041 | −0.0091343 | 0.0235170 | 0.29 |

**Notes.**

SE, standard error; CI, confidence intervals; DM, diabetes mellitus; HT, hypertension; TyG, Triglyceride glucose index.

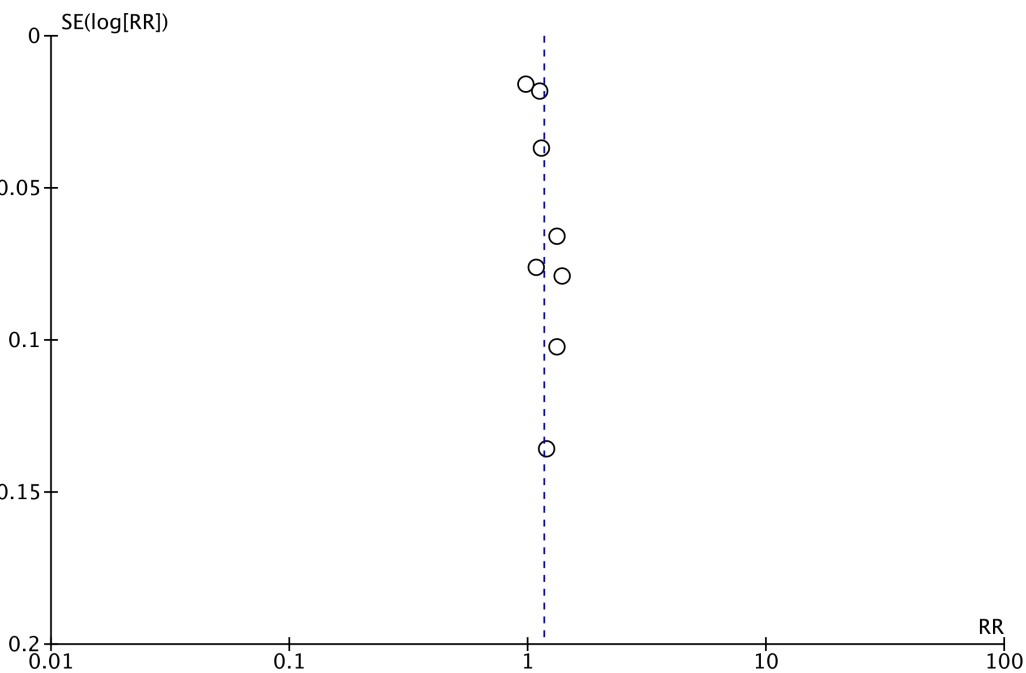

**Figure 5    Funnel plot for the meta-analysis with TyG as a continuous variable.**

During the search for an ideal marker, the TyG has generated considerable interest in the past few years. Its association with a large number of diseases as well as disease outcomes has prompted significant research on its ability to predict stroke in the general population (*Yin et al., 2024*; *Nayak et al., 2024*). However, despite three prior reviews (*Liao et al., 2022*; *Feng et al., 2022*; *Yang et al., 2023*), there remains uncertainty about its clinical application. One major limitation is the quality of these meta-analysis studies. *Yang et al. (2023)* in their review have not only included overlapping studies using the same dataset (Kailuan community cohort) but also have combined data of cross-sectional and cohort studies in the same meta-analysis. Repeated inclusion of the same data can overestimate the effect size generating skewed results. Also, cross-sectional studies cannot establish causality like cohort studies. Secondly, all three reviews (*Liao et al., 2022*; *Feng et al., 2022*; *Yang et al., 2023*) have also pooled studies assessing the risk of stroke in the general population with studies on specific disease cohorts like DM, HT, CAD, *etc*. The risk of stroke in the general population is considerably different compared to those with known risk factors like DM, HT, and CAD (*Saini, Guada & Yavagal, 2021*). Lastly, these reviews have not uniformly excluded studies including participants with prior stroke, which is another risk factor for recurrent stroke.

Overcoming these issues, the current systematic review and meta-analysis aimed to generate the best possible evidence on the clinical utility of TyG in predicting the risk of stroke in the general population. After the exclusion of a large number of studies with overlapping data, studies on disease populations, studies with short follow-ups, and studies including prior stroke participants, we were still left with 13 cohorts with about 12.8 million participants. Separate data analysis was conducted for TyG as a categorical as well as continuous variable. Our results revealed that high levels of TyG resulted in a 27% increase in the risk of stroke in the general population whereas a per unit increase in TyG was associated with a 16% increase of the same. We found the results to be robust on sensitivity analysis highlighting the credibility of the outcomes. Lack of publication bias also supplements the results. Our results are in agreement with the large study of *Lopez-Jaramillo et al. (2023)* which could not be included in the review as it did not exclude prior stroke patients. However, the study is worth mentioning as it was a prospective analysis of 141,243 individuals aged 35–70 years from 22 countries. After a median follow-up of 13.2 years, the authors reported an increased risk of stroke (hazard ratio: 1.16 95% CI [1.05–1.28]) with the highest TyG tertile. The association was the strongest in low-income countries followed by middle-income countries but non-significant in high-income countries.

Despite the robust results of our study, the small increase in the risk of stroke associated with high TyG may raise questions about its routine use in clinical practice. In the realm of stroke prevention, especially within the general population, establishing a precise minimal clinically important difference (MCID) is challenging due to varying baseline risks and individual patient factors. MCID represents the smallest change in the risk of disease or treatment outcome that a patient or clinician would identify as meaningful (*Salas Apaza et al., 2021*). Nevertheless, the perception of what constitutes a meaningful risk can be gauged from current CVD and stroke guidelines. For example, the American College of Cardiology and American Heart Association recommend considering statin therapy for

individuals with a 10-year atherosclerotic cardiovascular disease risk of 7.5% or higher (*Virani, 2022*). The risk of stroke noted with high TyG in our review was much higher and therefore should be clinically relevant and prompt monitoring and risk reduction measures in high-risk individuals.

An important caveat in understanding the utility of a marker for stroke prediction is the role of confounding factors. The risk of stroke depends on numerous variables like age, gender, ethnicity, family history, physical inactivity, alcohol consumption, smoking, obesity, DM, HT, CVD, *etc* (*Tu et al., 2023*). Much variation was observed in the included populations in terms of such baseline characteristics which could have led to the high heterogeneity in the meta-analysis. However, we were able to somewhat circumvent this limitation by including only adjusted data and conducting several subgroup and meta-regression analyses. Most of the included studies adjusted their data for age, gender, smoking, physical inactivity, body mass index, DM, HT, and alcohol consumption thereby eliminating the impact of major confounders. On subgroup analysis, there was no change in the significance of the results especially for TyG as a categorical variable. Segregation of data based on the region of the study, exclusion of CVD or stroke, diagnosis of stroke, TyG data as tertiles or quartiles, and follow-up had no impact albeit without complete elimination of inter-study heterogeneity. This indicates that there may be other unmeasured factors at play requiring further research. Likewise, meta-regression using relevant moderators like age, male gender, DM, HT, TyG cut-off, stroke incidence, and follow-up also did not have a significant association with the pooled analysis. The only significant association noted was between sample size and TyG as a continuous variable which could have been a statistical artifact given the small number of studies in the analysis.

The underlying mechanism supporting the link between high TyG and stroke remains unclear. Substantial evidence reinforces the relationship between the TyG index and insulin resistance (*Guerrero-Romero et al., 2010*; *Sánchez-García et al., 2020*). In fact, the TyG index has been identified as one of the best markers of insulin resistance performing better than visceral adiposity indicators and other lipid parameters (*Du et al., 2014*). Insulin resistance has been linked with stroke *via* several mechanisms like interference with insulin signaling and sensitivity, amplification of chronic systemic inflammation, and accelerating foam cell generation causing atherosclerosis and advanced plaques (*Bornfeldt & Tabas, 2011*; *Kosmas et al., 2023*). Insulin resistance interferes with the function of insulin-like growth factors, cyclic guanosine monophosphate, and nitric oxide thereby causing adhesion, activation, and aggregation of platelet function which in turn causes vascular occlusion and stroke (*Randriamboavonjy & Fleming, 2009*; *Guo et al., 2021*). Moreover, insulin resistance may impact the cerebrovascular reserve *via* several chemical, neuronal, and metabolic mechanisms leading to reduced cerebral perfusion during stroke (*Banks & Rhea, 2021*; *Fan et al., 2022*). There is also evidence to show that admission hyperglycemia and DM can negatively impact stroke outcomes itself (*Belge Bilgin et al., 2025*).

Despite the robust results, there are certain major limitations of the review. Firstly, we were unable to identify the ideal cut-off of TyG to predict the risk of stroke due to a lack of sensitivity and specificity data and varied cut-offs of the included studies. *Guerrero-Romero et al. (2010)* have shown that the best TyG cut-off for the diagnosis of insulin resistance

was 4.68. However, there remains limited data on the ideal cut-off for predicting stroke. Secondly, the studies considered only baseline glucose and triglyceride measurements and did not account for changes over time. It remains unclear how TyG changes affected the risk of stroke. Thirdly, despite the studies adjusting several potential cardiovascular and metabolic risk factors, we cannot eliminate the possibility of residual confounding due to unmeasured factors. Fourthly, data was derived from observational studies which have inherent bias. Fifthly, the large heterogeneity in the meta-analysis cannot be ignored and hence the results must be interested with caution. Lastly, the predominance of Chinese studies limits the generalization of data to other regions. More robust studies from Western countries are needed to add to current evidence.

## CONCLUSIONS

High TyG is associated with increased risk of stroke in the general population. Since the index is easy to measure and calculated from routinely available laboratory values, it may be incorporated into daily clinical practice to screen individuals at high risk of stroke.

### Funding
The authors received no funding for this work.

### Competing Interests
The authors declare there are no competing interests.

### Author Contributions
- Gang Xin conceived and designed the experiments, authored or reviewed drafts of the article, and approved the final draft.
- Huiya Li conceived and designed the experiments, performed the experiments, analyzed the data, prepared figures and/or tables, authored or reviewed drafts of the article, and approved the final draft.
- Ji Jiang conceived and designed the experiments, performed the experiments, analyzed the data, prepared figures and/or tables, and approved the final draft.

### Data Availability
  This is a systematic review/meta-analysis.

### Supplemental Information
Supplemental information for this article can be found online at http://dx.doi.org/10.7717/peerj.19994#supplemental-information.

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
