# Peer review of "Can the triglyceride-glucose index predict the risk of stroke? A meta-analysis of high-quality studies with 12.8 million participants"

_PeerJ, doi:10.7717/peerj.19994_

## Round 0.1 · original submission · Major Revisions

The reviewers have indicated various important points to be improved or corrected in the manuscript. Note that you revised manuscript will be send to a second round of review. To facilitate the review indicate in the rebuttal letter were the changes made to attend the reviewers comments appear.

Reviewer 1 ·

Basic reporting

- The manuscript is mostly well-written and comprehensible.

- The authors cite current and relevant literature on the TyG index and stroke risk, including recent meta-analyses. The introduction provides appropriate context for why TyG is a useful insulin-resistance marker associated with stroke risk​. However, one recent article is missing: “The effects of admission hyperglycemia and diabetes mellitus on mechanical thrombectomy outcomes: A systematic review and meta-analysis” (DOI: 10.1177/15910199241306774). Although focused on hyperglycemia and stroke treatment, it discusses glucose-related biomarkers in stroke and would strengthen the manuscript’s context on glycemic indices and cerebrovascular outcomes.

- The manuscript follows a standard structure, which is appropriate for PeerJ. The Abstract concisely states the objectives, methods, results, and conclusion. Each main section is clearly labeled. Figures and tables are generally well-placed.

Experimental design

- The study question is clearly defined and justified. The authors correctly identify a gap.

- The methods are generally detailed and appropriate. The authors searched multiple databases with clearly stated date limits, and they mention that the search strategy is detailed in a Supplementary Table. They also hand-searched references and Google Scholar. The review was registered on PROSPERO and follows PRISMA guidelines​, which is excellent practice.

- Data extraction methods are sound (dual extraction with conflict resolution). The authors reference the NOS for quality assessment in the references, implying they rated study quality. They report that all included studies scored 8–9 on NOS​, indicating high quality. It would be useful to explicitly describe the quality assessment process in Methods (criteria used, blinding of raters, etc.), if not already present. Ethical approval is not applicable (meta-analysis of published data) – this should be stated (often waived).

- The statistical approach is appropriate; however, the Methods should specify which statistical model was used (fixed vs. random effects – likely random given I²). Also, mention of any software (RevMan, Stata, etc.) and tests (e.g, Egger’s test) would be helpful. The text notes funnel asymmetry (Figure 3), but no formal bias test is reported – perhaps consider running Egger or trim-and-fill.

Validity of the findings

- The meta-analysis results support an association of higher TyG with greater stroke risk. These results are statistically significant, indicating a consistent positive association. It is appropriate that the authors conclude TyG is predictive of stroke risk. However, the magnitude of effect is moderate (RRs ~1.2–1.3). This should be interpreted with clinical context. The high I² values (66% and 89%) indicate notable heterogeneity, which tempers confidence. The authors did subgroup and meta-regression analyses. Notably, meta-regression found that larger sample sizes were associated with weaker associations, but none of the examined moderators (age, sex, baseline diabetes/hypertension prevalence, TyG cut-off, follow-up, etc.) significantly explained heterogeneity. This suggests robustness in the overall finding, but unexplained variability remains. The authors appropriately state that “Most results remained unchanged on subgroup analysis”.

- Each included study provided adjusted effect estimates (as per inclusion criteria), so the pooled results account for covariates to some extent. However, the exact covariates varied by study (likely including age, sex, BMI, blood pressure, lipids, etc.). The review should explicitly acknowledge this as a limitation: residual confounding cannot be ruled out.

- The funnel plot (Figure 3) reportedly indicates publication bias in the categorical TyG analysis. This potential bias (e.g., smaller studies showing stronger effects) should be more clearly discussed. Egger’s test could quantify this bias.

- The conclusion is supported by the pooled data. I suggest a slightly more cautious phrasing: for example, “Higher TyG is associated with increased stroke risk.” The current wording implies causality or high predictive power; it might be better to note that TyG shows a statistically significant association. The discussion should also mention limitations (observational data, heterogeneity, publication bias, residual confounding, and the fact that most studies were from East Asia, which may affect generalizability).

·

Basic reporting

The manuscript is a meta-analysis study to evaluate the ability of TyG to predict stroke in the general population, including only high-quality cohort studies. In this study, the authors conducted a thorough, comprehensive literature search of Embase, PubMed, CENTRAL, Web of Science, Scopus, and Google Scholar and finally included 13 studies. After information extraction and synthesis, they found that high levels of TyG resulted in a 27% increase in the risk of stroke in the general population, whereas a per-unit increase in TyG was associated with a 16% increase in the same. The writing language of the manuscript is clear and smooth, with appropriate logic. The result is persuasive, which implies TyG should be monitored and controlled for stroke prediction and prevention.

Experimental design

-

Validity of the findings

-

Additional comments

Several minor issues need to be mentioned:

1. Line 152-153 “….categorical and continuous variable separately in an inverse variance random-effects meta-analysis model.” Why was the random-effects meta-analysis model used, and why not the fixed-effects meta-analysis model? Can the authors explain?

2. It is suggested that some necessary optimizations be made to Figure 2 and Figure 4 to make the pictures clearer. The second and third columns of the forest plot are the calculated processing values before merging the effect sizes and do not need to be shown on the forest plot. It is recommended to delete them. The display range of the horizontal coordinate of the forest map can be narrowed (e.g., 0.5-5), making the result presentation more intuitive and allowing readers to clearly understand the content to be expressed. The content of column 6 is repetitive with that of column 1. It can be considered for deletion.

3. It is recommended that Table 2 be presented in the form of a forest map, which will be more intuitive and easier to understand

---

## Round 0.2 · accepted · Accept

Thank you for revising the text.

·

Basic reporting

The revised manuscript titled “Can the triglyceride-glucose index predict the risk of stroke? A meta-analysis of high-quality studies with 12.8 million participants” was a meta-analysis study to evaluating the ability of triglyceride-glucose (TyG) index to predict stroke in the general population including only high-quality cohort studies. In this study, the authors conducted a thoroughly comprehensive literature search of Embase, PubMed, CENTRAL, Web of Science, Scopus and Google Scholar and finally enclosed 13 studies. After information extraction and synthesis, they found that high levels of TyG resulted in a 27% increase in the risk of stroke in the general population whereas a per unit increase in TyG was associated with a 16% increase of the same.

The comments I have mentioned before have been completely revised and/or explained. I am extremely grateful to the author of the manuscript for their patient responses and meticulous revisions. I believe this manuscript has met the publication requirements of the journal and is acceptable for publication.

Experimental design

no comment

Validity of the findings

no comment

Additional comments

no comment